# X-PlugVid: Versatile Adaptation of Image Plugins for Controllable Video Generation

## Abstract

We introduce X-PlugVid, a unified framework designed to seamlessly adapt pre-trained image-based plug-and-play modules for video diffusion models, facilitating controllable video generation without the need for retraining. This framework leverages a spatial-temporal adapter to effectively bridge the gap between image and video diffusion models. Specifically, we adopt a frozen copy of a large-scale pretrained image diffusion model (*e.g.* Stable Diffusion v1.5) as spatial prior. Then we train a spatial-temporal adapter to convert the prior into temporally consistent guidance for video diffusion models (*e.g.* SVD). To further enhance the effectiveness of image plugins in guiding video models, we introduce a timestep remapping strategy. Recognizing that denoising is an entropic reduction process, this strategy selects priors from later timesteps of the image model, which contain richer information, to be injected into the video models, optimizing the quality and consistency of the generated videos. Comprehensive experimental evaluations of X-PlugVid demonstrate its broad compatibility with diverse operational conditions and different plugins, confirming that leveraging priors from a pretrained diffusion model can minimize redundant training and enable versatile, controllable video generation.

## 1 Introduction

Recent diffusion models have made significant advancements in generating high-quality images (Podell et al., 2023; StabilityAI; Midjourney; Li et al., 2024) and videos (Blattmann et al., 2023a; Zhang et al., 2023d;b; Wang et al., 2023b) from text descriptions. Despite these successes, controlling the structure and details of generated images using only text remains challenging. Therefore, many studies (Zhang et al., 2023c; Mou et al., 2023) have focused on controlling the image generation process by incorporating additional conditioning inputs such as bounding boxes (Li et al., 2023b), reference object images (Ruiz et al., 2023; Li et al., 2023a), and segmentation maps (Xie et al., 2023; Avrahami et al., 2023). These methods typically involve training a plug-and-play module, often referred to as plugins, on the basis of a large-scale pretrained image diffusion model to achieve conditioning. Inspired by these approaches, several studies (Zhang et al., 2023e; Chen et al., 2023b; Lin et al., 2024) attempt to replicate this success in video generation.

Similar to image generation, ControlVideo (Zhang et al., 2023e) proposes a training-free method to use image plugins for video generation, while Control-A-Video (Chen et al., 2023b) achieves the same goal by introducing an additional training-based temporal module. However, these methods have some issues: 1) they lack flexibility and cannot be easily transferred to different pretrained models. For instance, a plugin module developed for Control-A-Video cannot be readily transferred to other video models like SVD. 2) These approaches require extensive training data. To train each plugin, it's necessary to label specific video conditions such as video depth, sketches, and bounding boxes, which is more costly and time-consuming than that for images. In contrast, the image diffusion models already benefit from numerous effective plugins facilitating controlled image generation. This raises a question: ***can these plugins designed for images be effectively adapted for video generation models?***. Thus, in this work, we explore the application of image-based plugins to video models for video generation. Notably, we only focus on image plugins whose function is spatial control (*e.g.* ControlNet (Zhang et al., 2023c), T2I-Adapter (Mou et al., 2023)) in this work.

Recent research (Ran et al., 2024) has demonstrated that the domain gap between two different versions of image models can be bridged effectively with a well-designed adapter. This allows the upgraded model to be universally compatible with all plugins of the base model without the need for retraining. However, when extending this approach to bridge image and video models, we must also consider the modality gap. Compared to image data, videos represent higher-dimensional and more complex data distributions than images. Furthermore, by analyzing the principles of X-Adapter (Ran et al., 2024) and ControlNet (Zhang et al., 2023c), we found that the injection of high-frequency additional features at every timestep is essential for guidance while previous works overlook this point.

In this work, we propose X-PlugVid to equip the video diffusion model with pretrained image plugins for high-quality and consistent controllable video generation without the need for retraining. X-PlugVid universally allows all spatial-control plugins to function with video diffusion models by training a generic adapter. Unlike prior work (Ran et al., 2024), we extend the spatial adapter to a spatial-temporal adapter, enabling it to simultaneously possess domain adaptation and temporal modeling capabilities. Additionally, by analyzing the spectral characteristics of the adapter, we found that it tends to learn low-frequency components to generate smooth background and camera movements but lacks high-frequency features, often resulting in the loss of the subject's appearance. To address this issue, high-frequency pass filtering is applied to the adapter's input.

To enable better guidance, we further analyze the conditioning principles of ControlNet (Zhang et al., 2023c). Based on our findings, we propose a novel timestep remapping method. Specifically, we found that the features of the image diffusion model in the early steps contain insufficient information for effective guidance, though early timesteps are crucial for determining low-frequency components like layout. To improve guidance at these critical early timesteps, we have moved away from synchronously mapping the timesteps of the image and video model backbones. Instead, we strategically map the later timesteps of the image model, which contain richer information, to the earlier timesteps of the video model. This adjustment allows us to inject more useful information early in the video generation process, improving the overall quality and coherence of generated videos.

In our experiments, we first demonstrate that our method shows good generalizability across various types of video models, i.e., text-to-video models like Hotshot-XL (Mullan et al., 2023) and image-to-video models like (Zhang et al., 2023d). Next, we demonstrate that our approach shows strong compatibility with various types of image plugins and surpasses previous methods for controllable video generation. Lastly, we provide comprehensive ablations for the design choices of X-PlugVid and qualitative examples.

In summary, the contribution of this paper can be summarized as:

- We target a new task in the large-scale generative model era where we efficiently reuse pretrained image plugin for controllable video generation.

- We analyze the mechanism of utilizing pretrained diffusion models as spatial prior. Based on our findings, we design a spatial-temporal adapter for guidance and introduce a novel timestep remapping strategy to enhance the adapter's guidance ability.

- Experiments show the proposed method demonstrates compatibility with various conditions and plugins and surpasses previous methods in terms of performance.

## 2 RELATED WORKS

**Text-to-video and Image-to-video models**. The field of video generation has witnessed significant progress recently due to the advancement of diffusion models (Sohl-Dickstein et al., 2015; Dhariwal & Nichol, 2021) and large-scale text-video dataset (Chen et al., 2024b). These models generate videos from text descriptions or images. Imagen Video (Ho et al., 2022a) utilizes a cascading structure for high-resolution text-to-video generation while Video Diffusion Model (Ho et al., 2022b) expands the standard image architecture to accommodate video data and trains on both image and video together. Other methods develop video models based on powerful text-to-image models like Stable Diffusion (Rombach et al., 2021), adding extra layers to capture cross-frame motion and ensure consistency. Among these, Tune-A-Video (Wu et al., 2023) employs a causal attention module

and limits the training module to reduce computational costs. Align-Your-Latents (Blattmann et al., 2023b) efficiently transforms T2I models into video generators by aligning independently sampled noise maps. AnimateDiff (Guo et al., 2024) utilizes a plug-and-play temporal module to enable video generation on personalized image models (StabilityAI). Other text-to-video works include marrying latent and pixel space (Zhang et al., 2023b) and cascaded genration (Wang et al., 2023b). To address high-quality video generation tasks, several works (Zhang et al., 2023d; Chen et al., 2023a; 2024a) develop image-to-video models and all of them achieve remarkable pixel quality.

**Controllable video generation**. Since text prompts often provide unclear guidance regarding the motions and spatial structure of videos, making such control mechanisms is essential in video generation. For high-level control over video motion, some work proposes to use motion trajectories (Yin et al., 2023), pose sequences (Ma et al., 2024) while some work uses Low-Rank Adaptations(LoRA) (Hu et al., 2021) to learn specific motion patterns (Zhao et al., 2023b). For fine-grained spatial structure control, Gen-1 (Esser et al., 2023) first introduces the use of depth sequences as guidance. VideoComposer (Wang et al., 2023a) incorporate several conditions during training while other methods (Chen et al., 2023b; Zhang et al., 2023e) adopt pretrained image ControlNet (Zhang et al., 2023c) for video generation. Though these methods achieve fine-grained controllability, they often require a substantial amount of computational resources for training. We aim to reduce the required computational resources by efficiently reusing pretrained image plugins.

**Parameter-Efficient Transfer Learning**. Our task is related to parameter-efficient transfer learning as well since our goal is to eliminate the domain and modality gap between image and video diffusion models. The emergence of large-scale pre-trained models like CLIP (Radford et al., 2021), Stable Diffuions (Rombach et al., 2021) has underscored the significance of effectively transferring these foundational models to downstream tasks. Parameter-efficient Transfer Learning (PETL) methods (Houlsby et al., 2019; Zhang et al., 2023a; Zhao et al., 2023a) introduce additional parameters to the original model to bridge the domain gaps between the pre-trained dataset and target tasks. X-Adapter (Ran et al., 2024) propose a spatial adapter to bridge diffusion models of different versions and enable plugins pretrained on old version (*e.g.* SD1.5 (StabilityAI)) to be directly applied to upgraded version(*e.g.* SDXL (Podell et al., 2023)). Similar to X-Adapter, our objective is to effectively reuse pretrained image plugin on video diffusion model.

## 3 METHOD

### 3.1 TASK DEFINITION

We aim to design a universally compatible adapter so that pretrained image plugins whose function is spatial control can be directly used in video diffusion model *without plugin-specific retraining*, as illustrated in Fig. 1(a). Typically, adapting image plugins to video models might involve retraining each plugin separately, as shown in Fig. 1(b). For instance, considering the ControlNet (Zhang et al., 2023c) family, which comprises over twenty distinctive plugins, such retraining would demand excessive and repetitive training efforts to maintain the original functionalities of each plugin. In contrast, our method only requires training a single backbone-to-backbone adapter. This allows for the seamless integration of all pretrained spatial-control plugins from the image model, significantly enhancing efficiency and reducing the resources required for adaptation.

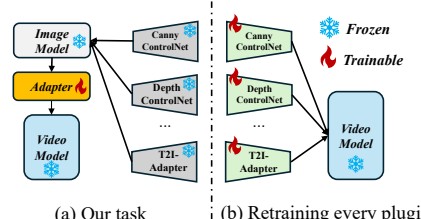

(a) Our task     (b) Retraining every plugin

Figure 1: *Task Definition*. Different from previous works, we train a single adapter to enable all spatial-control image plugins work with video models

### 3.2 PRELIMINARY

**Denoising Diffusion Probabilistic Models (Ho et al., 2020)** DDPMs are designed to capture the underlying data distribution by leveraging two mechanisms: diffusion and denoising. Starting with an input data sample $z \sim p(z)$, the diffusion process incrementally introduces noise into $z$ following formula $z_t = \alpha_t z + \sigma_t \epsilon$, where $\epsilon \sim \mathcal{N}(0, I)$. This process is a Markov chain consisting

of T steps, with the Signal-to-noise(SNR) ratio $\sigma_t^2/\alpha_t^2$ decreasing over time. Ideally, $z_T$ will follow pure Gaussian noise. In the denoising stage, a denoiser $\epsilon_\theta$ is employed to predict added noise $\epsilon$. Formally, $\epsilon_\theta$ is trained using the following objective:

$$\min_\theta E_{z,\epsilon\sim N(0,I),\boldsymbol{t}\sim \text{Uniform}\,(1,T)} \|\epsilon - \epsilon_\theta\left(\boldsymbol{z}_t,\boldsymbol{t}\right)\|_2^2, \tag{1}$$

**Latent diffusion models (Rombach et al., 2021)** LDM extends DDPMs by operating in the latent space. It leverages a pretrained VAE to compress the RGB image z to latent space using VAE's encoder $\varepsilon$. After adding noise to the latent, $\epsilon_\theta$ will denoise it iteratively. Formally, $\epsilon_\theta$ is trained using following formula:

$$\min_\theta E_{z,\epsilon\sim N(0,I),\boldsymbol{t}\sim \text{Uniform}\,(1,T)} \|\epsilon - \epsilon_\theta\left(\varepsilon(\boldsymbol{z}_t),\boldsymbol{t}\right)\|_2^2, \tag{2}$$

### 3.3 X-PLUGVID

We introduce a novel framework termed X-PlugVid, which effectively reuses image plugins for video controllable generation. The core of our method is based on two crucial insights on controllable generation and universal adaptation. By analyzing the mechanisms of ControlNet (Zhang et al., 2023c) and X-Adapter (Ran et al., 2024), we found that: 1) The key to controllable generation is the injection of high-frequency control patterns at every desnoising step. 2) Pretrained image diffusion models can serve as priors for spatial control since their feature maps contain the necessary patterns. These patterns can be manipulated by plugins and transferred by adapters, which is the reason for X-Adapter (Ran et al., 2024)'s universal adaptation. Based on our findings, we propose a spatial-temporal adapter to bridge the image and video models. Additionally, a high-pass filter is applied to the adapter's input to ensure high-frequency components are adapted seamlessly. Finally, we propose a novel timestep remapping strategy to provide better guidance at early timesteps.

#### 3.3.1 HOW DO CONTROLNET AND X-ADAPTER WORK?

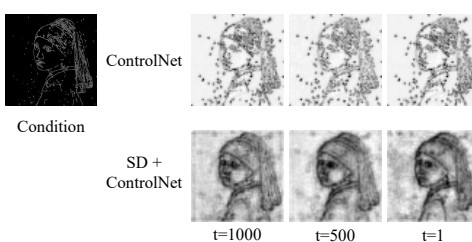

Figure 2: Visualization of feature maps of ControlNet and Stable Diffusion. The diffusion model's feature maps and ControlNet's outputs exhibit high similarity.

Figure 3: Frequency Characteristics of Stable Diffusion *w.* and *w.o.* ControlNet. Compared to the diffusion model, the high-frequency components dominate the output of ControlNet

We first investigate the mechanisms behind the success of ControlNet (Zhang et al., 2023c) and X-Adapter (Ran et al., 2024)

**How Does ControlNet work?** ControlNet copies the encoder of the backbone, which is always a UNet (Ronneberger et al., 2015), takes the condition as input and adds its output to the backbone's decoder. To visualize ControlNet's output feature map, we first compute the average feature map along the channel dimension and normalize it to $[0, 1]$. As depicted in Fig. 2, ControlNet generates the condition pattern at every timestep and injects them into the backbone as guidance. Moreover, by analyzing the frequency characteristics of ControlNet's outputs as shown in Fig. 3, we find that ControlNet mainly produces high-frequency patterns. Based on these findings, we conclude that ControlNet's primary mechanism is injecting high-frequency condition patterns into the backbone at every timestep.

**How Does X-Adapter work?** X-Adapter (Ran et al., 2024) discovers that after applying ControlNet to a diffusion model, the model's feature maps and ControlNet's outputs exhibit similarity, as shown in Fig. 2. Based on this finding, X-Adapter takes diffusion models's feature maps as prior and

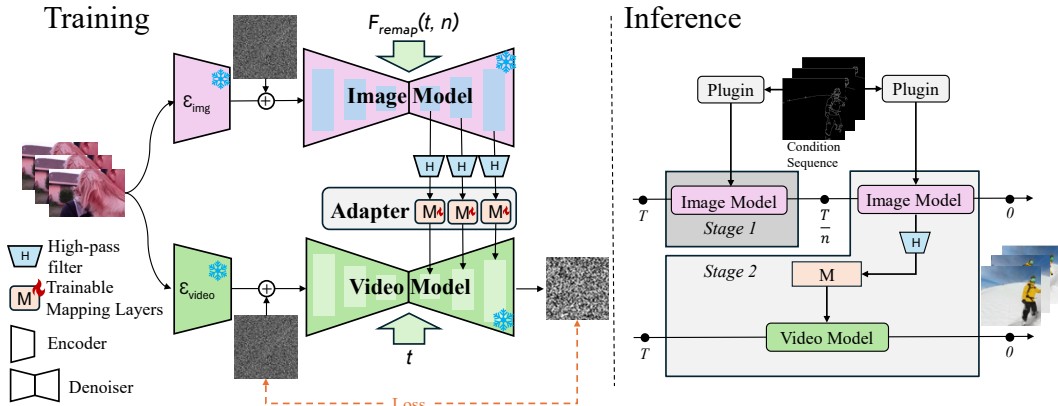

Figure 4: ***Method Overview***. In training, different noises are added to image and video model in the latent domain. After sampling timestep $t$ for video model, we get corresponding image model timestep using remapping function $F_{remap}$. Note that no plugin is involved during training. In inference, the denoising process is divided into two stages. In the first stage, only image model runs until it reaches timestep $\frac{T}{n}$. In the second stage, two backbones inference together under remapped timesteps.

transfers them from the base backbone's space to the upgraded backbone's space while preserving their original patterns. Since these adapted features already contain the necessary condition patterns, they can serve as guidance for the upgraded backbone. Additionally, plugins like ControlNet or T2I-Adapter (Mou et al., 2023) can also control the generation process through the base model. This method demonstrates that pretrained backbones can be utilized as spatial prior, enabling universal adaptation.

However, one important aspect X-Adapter overlooks is that, unlike ControlNet, the feature map patterns of the backbone contain redundant low-quality components and are subtle at early timesteps as shown in Fig. 2 and Fig. 3. Applying spatial priors to video models also remains unexplored. Therefore, our work mainly focuses on how to better utilize the spatial priors of pretrained diffusion models.

### 3.3.2 SPATIAL-TEMPORAL ADAPTER

Based on our analysis in Sec. 3.3.1, though the image diffusion model can be used as spatial prior, it lacks temporal modeling, which makes it challenging to be directly applied to video diffusion models. To overcome this domain gap, we add a temporal module to our adapter. In detail, X-PlugVid is built upon Stable Diffusion v1.5 (StabilityAI) to ensure compatibility with the plugins' connectors. Within certain decoder layers, extra mapping networks are added and trained. These mapping layers are referred to as the adapter in this paper. The adapter's function is to map features from the space of the image model to the video model (e.g., SVD (Blattmann et al., 2023a)) for guidance. Since the features from the image model are temporally inconsistent, directly utilizing these adapted features as guidance would result in the degradation of video quality and consistency. Therefore, we introduce a temporal attention (Vaswani et al., 2017) module to ensure temporal coherence.

Additionally, we discovered that the feature maps of the image model contain abundant low-frequency information. According to our analysis in Sec. 3.3.1, ControlNet's outputs maintain at high frequency across all timesteps, whereas diffusion spatial prior *i.e.* feature maps, does not meet this condition. Moreover, our experiments found that these components often adversely affect the adapter's guidance since they always contain low-quality parts as shown in Sec. 4.4. Thus, we apply high-pass filtering to the image features before feeding them to the adapter to filter these components. Formally, suppose we have $N$ adapters and $\mathcal{M}_n(\cdot)$ denotes the $n^{th}$ trained mapper, given multi-scale feature maps $\boldsymbol{F}_{img} = \{\boldsymbol{F}_{img}^1, \boldsymbol{F}_{img}^2, ..., \boldsymbol{F}_{img}^N\}$ from image model, the guidance feature fusion can be defined as the following formulation:

$$\boldsymbol{F}_{video}^n = \boldsymbol{F}_{video}^n + \mathcal{M}_n(\mathcal{H}(\boldsymbol{F}_{img}^n)), n \in \{1, 2, ..., N\} \tag{3}$$

where $\mathcal{H}()$ is a high-pass filter, $\boldsymbol{F}_{video}^{n}$ denotes video model's $n^{th}$ decoder layer to fuse guidance feature.

### 3.3.3 TIMESTEP REMAPPING

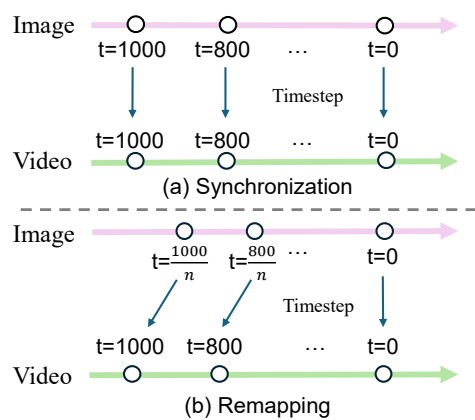

Figure 5: Timestep Remapping. (a) X-Adapter adopts synchronous timesteps. (b) We map later timesteps from the image model to earlier steps of the video model to provide sufficient guidance in the early steps.

Figure 6: Denoising Trajectory of video model under various settings. With timestep remapping, the adapter provides sufficient guidance during the early timesteps, allowing its trajectory to deviate from the original I2Vgen-XL trajectory and achieve better results

Our temporal adapter now possesses the capability to convert the priors of diffusion models into guidance. According to our analysis, it is also necessary to output stable patterns at each timestep. However, the priors from the image diffusion model do not contain sufficient information in the early timesteps. Directly applying it would result in lack of guidance in early timesteps. To confirm this issue, we visualize the denoising trajectories of the video model with and without the involvement of the adapter. By performing Principal Component Analysis (PCA) on denoising results at each step and taking the first two components, we plot the denoising trajectory as shown in Fig. 6. It is obvious that the adapter's guidance is too subtle in the early stages, leading to suboptimal results shown in Fig. 8. To address this issue, we propose a novel method denoted as timestep remapping.

Generally, the backward progress of two backbones are synchronized throughout the entire denoising process (Ran et al., 2024). It means that the timesteps of two backbones are always synchronized and put in a one-to-one correspondence during training and inference. In our method, the timesteps are no longer synchronized; later timesteps from the image model are mapped to earlier steps of the video model as shown in Fig. 5. The motivation is that later timesteps of the image model contain more useful information. Injecting this information into the early timesteps of video model can enhance the guiding capability of the adapter. Formally, given timestep $t_{vid}$ of video model, its corresponding image model timestep $t_{img}$ is computed using following formulation:

$$t_{img} \quad = \quad F_{remap}(t_{vid}, n) \tag{4}$$

$$F_{remap}(t_{vid}, n) \quad = \quad \left\lceil \frac{t_{vid}}{n} \right\rceil, n \in \{1, 2, 3..., T\} \tag{5}$$

where $n$ is the hyperparameter for timestep remapping, $T$ is the total number of timesteps, $F_{remap}$ is the remapping function. When $n = 1$, timestep remapping is equivalent to timestep synchronization. We observe that $n = 2$ is suitable in most cases. We give detailed ablations on timestep remapping strategy in the experiments Sec. 4.4.

### 3.3.4 TRAINING AND INFERENCE

As shown in Fig. 4, given a video diffusion model, X-PlugVid is trained in a **plugin-free** manner for video generation. Similar to X-Adapter (Ran et al., 2024), to ensure that image plugins can be seamlessly inserted, we fix two backbones' parameters and only update the temporal-spatial adapter from scratch. We use the same training objective in standard LDMs (Rombach et al., 2021).

Formally, given input video frames $\mathcal{V}$, we first embed it to the latent spaces $z_0$ and $\overline{z}_0$ via image and video autoencoder, respectively. Then, we randomly sample a timestep $t_{vid}$ of video model, and adopt timestep remapping as introduced in Sec. 3.3.3 to get image model's timestep $t_{img}$. After that, we add noise to the latent space, and produce two noisy latent $z_t$ and $\overline{z}_t$ for denoising. X-PlugVid is trained with the video diffusion network $\epsilon_\theta$ to predict the added noise $\epsilon$ by:

$$E_{\overline{z}_0, \epsilon, t_{vid}} \left\| \epsilon - \epsilon_\theta \left( z_t, t_{vid}, \overline{z}_t, t_{img} \right) \right\|_2^2. \tag{6}$$

After training, the plugins of image models can naturally be added for their abilities.

During inference, to align with the remapping strategy in training, the inference process is divided into two stages. In the first stage, the image model runs independently until timestep $\frac{T}{n}$, where $n$ is the remapping hyperparameter. In the second stage, both backbones perform inference simultaneously and we ensure that at each step $t_{img}$ and $t_{vid}$, the timesteps of two backbones, always satisfy $t_{img} = F_{remap}(t_{vid}, n)$.

## 4 EXPERIMENTS

### 4.1 IMPLEMENTATION DETAILS

We implement X-PlugVid using Stable Diffusion v1.5 (StabilityAI) as the image model, I2VGen-XL (Zhang et al., 2023d) and Hotshot-XL (Mullan et al., 2023) as the main video model. Notice that we also train our method for SVD (Blattmann et al., 2023a), which shows promising results as shown in the appendix. The adapter of X-PlugVid is placed at the image model's middle block and the first three decoder blocks, containing four mapping layers. For training, we randomly sample a subset of Panda70M (Chen et al., 2024b) training set containing 100k text-video pairs for training. We utilize the AdamW optimizer with a learning rate of $1e^{-5}$ and a batch size of 8. The model is trained for 5 epochs using 4 NVIDIA A100 GPUs.

### 4.2 QUALITATIVE RESULT

As depicted in Fig. 7, we show the qualitative results of our method on both I2VGen-XL (Zhang et al., 2023d) and Hotshot-XL (Mullan et al., 2023) using different conditions, which demonstrates compatibility across various conditions. Additionally, our method is also compatible with other plugins besides ControlNet, such as T2I adapter (Mou et al., 2023). We also provide video results in the appendix.

### 4.3 COMPARISONS

Table 1: Comparison of various methods on depth map and canny edge.

| Method | Depth Map | | Canny Edge | |
|---|---|---|---|---|
| | FID ($\downarrow$) | Optical Flow Error ($\downarrow$) | FID ($\downarrow$) | Optical Flow Error ($\downarrow$) |
| ControlVideo (Zhang et al., 2023e) | 39.16 | 6.27 | 39.78 | 6.21 |
| Control-A-Video (Chen et al., 2023b) | 35.14 | 5.02 | 36.01 | 4.81 |
| VideoComposer(Wang et al., 2023a) | 33.24 | 5.72 | - | - |
| Hotshot-XL + X-PlugVid (Ours) | 30.62 | 3.49 | 30.84 | 3.12 |
| I2VGen-XL + X-PlugVid (Ours) | **29.21** | **3.31** | **29.63** | **3.02** |
| **Performance of original video model** | | | | |
| I2VGen-XL(Zhang et al., 2023d) | 29.09 | - | 29.09 | - |
| Hotshot-XL(Mullan et al., 2023) | 30.51 | - | 30.51 | - |

**Experiment setting.** We conduct experiments using X-PlugVid on text-to-video generation backbone Hotshot-XL (Mullan et al., 2023) as well as image-to-video generation backbones I2VGen-XL (Zhang et al., 2023d). We choose two representative spatial-control plugins, canny and depth controlnet (Zhang et al., 2023c) to evaluate the performance of the proposed method. These two kinds of controlnet represent dense and sparse conditions seperately, which covers most cases in controllable generation. We utilize the Panda70M (Chen et al., 2024b) validation set, which contains

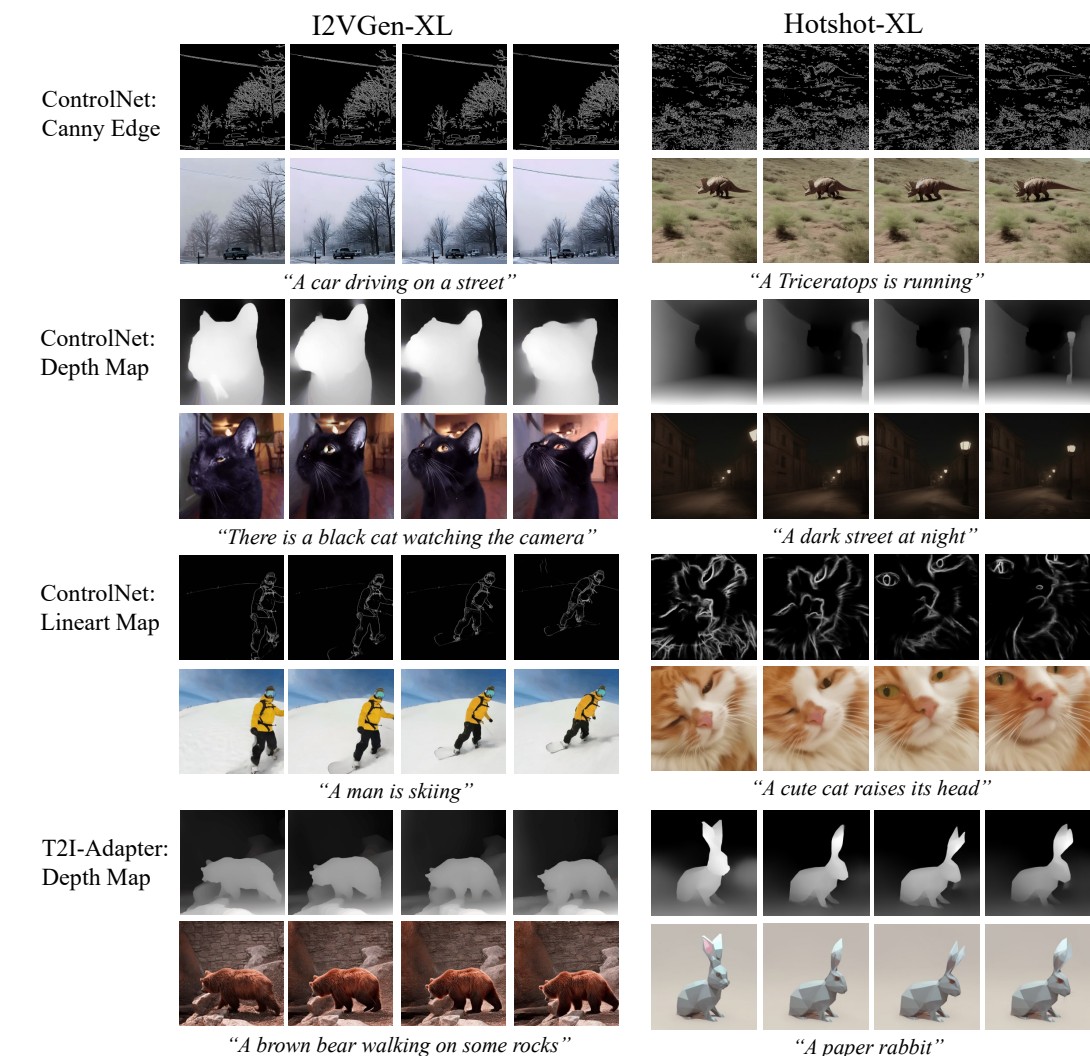

Figure 7: *Qualitative Result*. Our X-PlugVid successfully adapts image plugins (ControlNet (Zhang et al., 2023c) and T2I-Adapter (Mou et al., 2023)) to video models (I2VGen-XL (Zhang et al., 2023d) and Hotshot-XL (Mullan et al., 2023)) and exhibits compatibility with different conditions and plugins.

2000 text-video pairs, to evaluate each method. We compare our method with previous controllable video generation models, Control-A-Video (Chen et al., 2023b), ControlVideo (Zhang et al., 2023e), and VideoComposer (Wang et al., 2023a). Notably, we also compare our method with original backbones, *i.e.* Hotshot-XL and I2VGen-XL to demonstrate that our method does not harm their native generation capabilities.

As for evaluation metrics, we use Frechet Inception Distance (FID) to measure the distribution distance between videos generated by our method and the original videos, which indicates video quality. Following VideoControlNet (Hu & Xu, 2023), we also calculate the L2 distance between the optical flow (Ranjan & Black, 2016) of the input video and the generated video, namely optical flow error. **Compare to other methods.** Table. 1 demonstrates that, under both depth map and canny edge conditions, X-PlugVid on I2VGen-XL and Hotshot-XL surpass all previous video control methods in terms of visual quality (FID) and spatial control (optical flow error) metrics. Meanwhile, our method achieves FID scores comparable to original I2VGen-XL and Hotshot-XL. This indicates that our method not only extends the functionality of the original model but also retains its generative capabilities flawlessly.

## 4.4 ABLATIVE STUDY

Table 2: Ablation Study

| | Depth Map | | Canny Edge | |
|---|---|---|---|---|
| | **FID** ($\downarrow$) | **Optical Flow Error** ($\downarrow$) | **FID** ($\downarrow$) | **Optical Flow Error** ($\downarrow$) |
| **Ablation on High-pass Filter and Timestep Remapping** | | | | |
| *w.o.* High-pass Filter & Timestep Remapping | 36.22 | 7.25 | 36.16 | 7.01 |
| *w.o.* Timestep remapping | 32.64 | 6.09 | 32.88 | 5.92 |
| *w.o.* High-pass Filter | 30.64 | 4.71 | 30.76 | 4.33 |
| Full Method | **29.21** | **3.31** | **29.63** | **3.02** |
| **Ablation on Mapping Layer Insertion** | | | | |
| Encoder | 31.06 | 4.56 | 31.23 | 4.32 |
| Decoder | **29.21** | **3.31** | **29.63** | **3.02** |
| **Ablation on $n$ in Timestep Remapping** | | | | |
| $n = 1$ | 32.64 | 6.09 | 32.88 | 5.92 |
| $n = 2$ | **29.21** | **3.31** | **29.63** | **3.02** |
| $n = 4$ | 33.21 | 3.42 | 33.54 | 3.09 |
| $n = 1000$ | 34.14 | 3.92 | 34.38 | 3.76 |

Our ablation study is based on I2VGen-XL. We mainly focus on three questions:

**Where to insert the mapping layers?** We study the effect of inserting mapping layers into different modules: (1) Encoder; (2) Decoder. Table. 2 indicates that inserting mapping layers to decoder shows strongest guidance capability since it retains the feature space of encoder.

**How important are the high-pass filter and timestep remapping?** To demonstrate the effectiveness of high-pass filter and timestep remapping, we conduct a comparison with the following variants: i) No high-pass filter and timestep remapping ii) high-pass filter only iii) timestep remapping only iv) our full method. Note that, all the above experiments are based on temporal adapter. The quantitative results are shown in Table. 2. The result indicates that timestep remapping significantly enhances adapter's guidance capability, making the generated results more align with the conditions. The complement of the high-pass filter further eliminates unnecessary information from the diffusion prior *i.e.* Stable Diffusion v1.5 (StabilityAI), preventing low-quality priors affecting the original video model's generative ability, thus improving image quality and consistency.

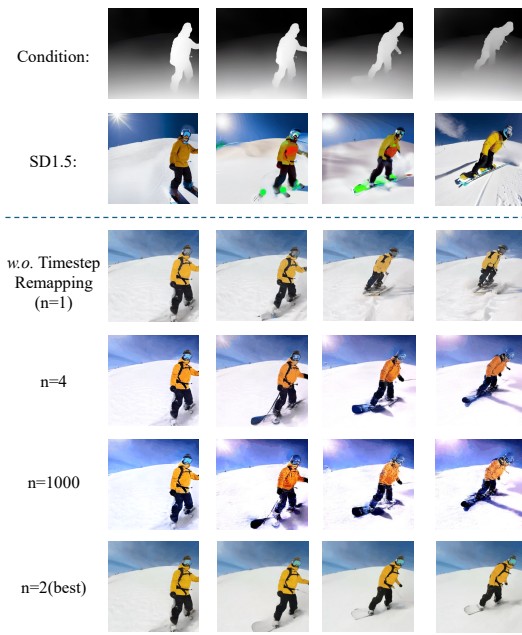

Figure 8: Ablation on the effect of $n$ in timestep remapping.

**What is the effect of $n$ in timestep remapping?** We conduct experiments with four different values of $n$: 1, 2, 4, and 1000. When $n = 1$, timestep remapping strategy is equivalent to timestep synchronization. When $n = 1000$, which is the maximum value it can reach, we map the feature of the image model's last timestep to all timesteps of the video model. As depicted in Table. 2 and Fig. 8, the results show that when $n = 1$, the guidance is too weak to align the generated results with the conditions. We visualize the output of the adapter with and without timestep remapping as shown in Fig. 9. It shows that without timestep remapping, the adapter's output becomes blurry, and the inconsistency between different frames increases, leading to flickers. This is because the features at early timesteps contain less semantic information, have greater uncertainty, and are more difficult for the adapter to learn temporal consistency. When $n = 2$, the result significantly improves, but as we continue to increase its value, the video quality largely degrades. Our generated results, in

terms of overall color tone and details like clothing, become increasingly similar to SD1.5 with the increase of $n$ as shown in Fig. 8. This is because the guidance gets stronger as $n$ increases, and low-quality components of the prior *i.e.* stable diffusion 1.5 (StabilityAI) are transferred by the adapter and degrade the final generation quality. It reveals that if the value of $n$ is too high, we cannot completely eliminate all low-quality parts even with our adapter and high-pass filter. In conclusion, we achieve the best results only when the guidance strength is appropriate, specifically at $n = 2$.

# 5    DISCUSSION

## 5.1    GENERALIZATION

Although we mainly discuss how to use diffusion prior in controllable video generation, the strategies we propose: applying high-pass filter and timestep remapping, are not task-specific. To verify the generality of our method, we apply them to image model upgrade, which is the same as X-Adapter (Ran et al., 2024)'s task. Specifically, we implement timestep remapping and high-pass filter upon X-Adapter and achieve better results. Please refer to our appendix for qualitative and quantitative results.

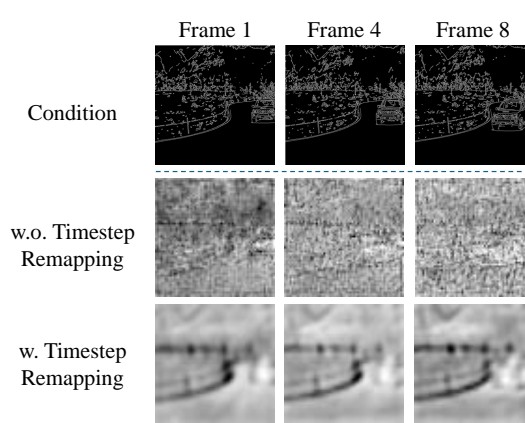

Figure 9: Visualization of adapter's output *w.* and *w.o.* Timestep Remapping.

In addition to controllable video generation, our method is also applicable to video editing as shown in Fig. 10. By using spatial conditions extracted from the original video along with the target prompt, our method can generate a high-quality video that aligns with the target text while preserving the spatial layout and dynamics of the input video.

## 5.2    LIMITATIONS

In this work, we focus on how to better utilize the spatial information in the diffusion prior, enabling us to reuse spatial-control image plugins in the video diffusion model. However, the diffusion prior also contains other information, such as identity and style. If we can better leverage this information, plugins like LoRA (Hu et al., 2021) and IP-Adapter (Ye et al., 2023) can be applied to the video model as well. We leave these capabilities as future work.

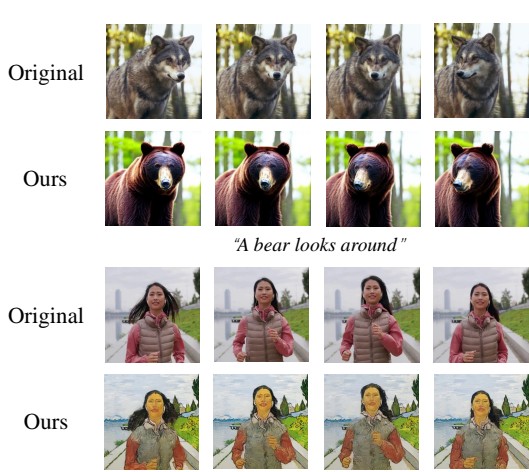

Figure 10: Qualitative results on video editing.

# 6    CONCLUSION

In this paper, we target a new task of reusing image plugins for controllable video generation. To this end, we analyze how to adopt pre-trained diffusion model as spatial prior. Based on our findings, we design a spatial-temporal adapter for guidance and apply high-pass filter to the input of adapter to filter low-quality components. To enhance adapter's guidance ability, we design a novel timestep remapping strategy to insert fine-grained information to video diffuson model. We conduct comprehensive experiments to demonstrate the advantages of the proposed methods.

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

# A    DETAILED NERWORK STRUCTURE

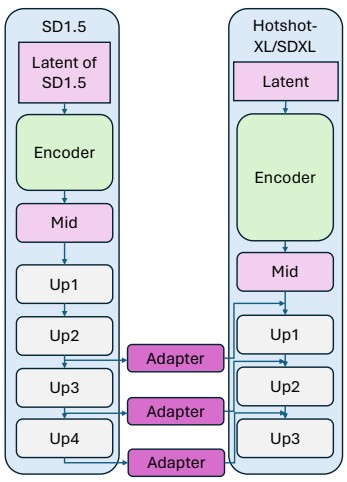

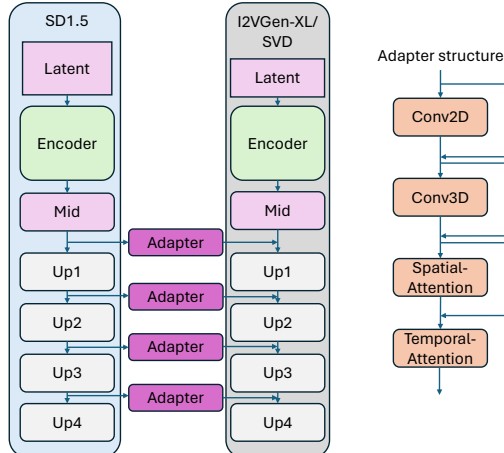

Figure 1: The location adapter insert when train adapter on Hotshot-XL and SDXL.

Figure 2: Network structure of adapter and the location adapter insert when train with I2VGen-XL and SVD.

The network architecture of adapter when we train adapter with I2VGen-XL (Zhang et al., 2023d), SVD (Blattmann et al., 2023a), Hotshot-XL (Mullan et al., 2023) and SDXL (Podell et al., 2023) are shown in Fig. 2 and Fig. 1.

# B    QUALITATIVE VIDEO RESULTS

As depicted in Table. 2, we show the qualitative results of our method on SVD using different conditions, which demonstrates compatibility across various conditions.

# C    QUALITATIVE AND QUANTITATIVE RESULTS ON IMAGE MODEL

Table 1: Quantitative evaluation against X-Adapter.

| Plugin: **ControlNet** | FID ↓ | CLIP-score ↑ | Cond. Recon. ↑ |
|---|---|---|---|
| X-Adapter | 30.95 | 0.2632 | 0.27 ± 0.13 |
| X-Adapter + X-PlugVid | **30.89** | **0.2643** | **0.32 ± 0.11** |

To demonstrate the generalization of our method, we apply high-pass filter and implement timestep remapping based upon X-Adapter (Ran et al., 2024). For training, we randomly sample a subset of f Laion-high-resolution containing 300k text-image pairs for training to align with training setting of X-Adapter. We utilize the AdamW optimizer with a learning rate of $1e^{-5}$ and a batch size of 8. The model is trained for 2 epochs using 4 NVIDIA A100 GPUs. The evaluation setting follows X-Adapter(Ran et al., 2024). The qualitative and quantitative results are shown in Fig. 3 and Table. 1. The results shows that with the help of our method, X-Adapter achieves better condition fidelity, demonstrating the generality of our approach in better leveraging the spatial priors of diffusion models.

Table 2: Qualitative video results (Best viewed with a PDF reader that supports GIF display and click to play videos).

Some ducks are playing in the pond.

A man with a black hat is smiling.

A snail is crawling slowly.

An old woman with silver hair and glasses.

Close up of an astronaut's face.

A teddy bearsurrounded by children's toys.

A burning candle.

A cute cat outside the window.

Target prompt: A bear is looking around.

Target prompt: A woman is running, Van Gogh style.

| Condition | X-Adapter | X-Adapter + X-PlugVid | Condition | X-Adapter | X-Adapter + X-PlugVid |

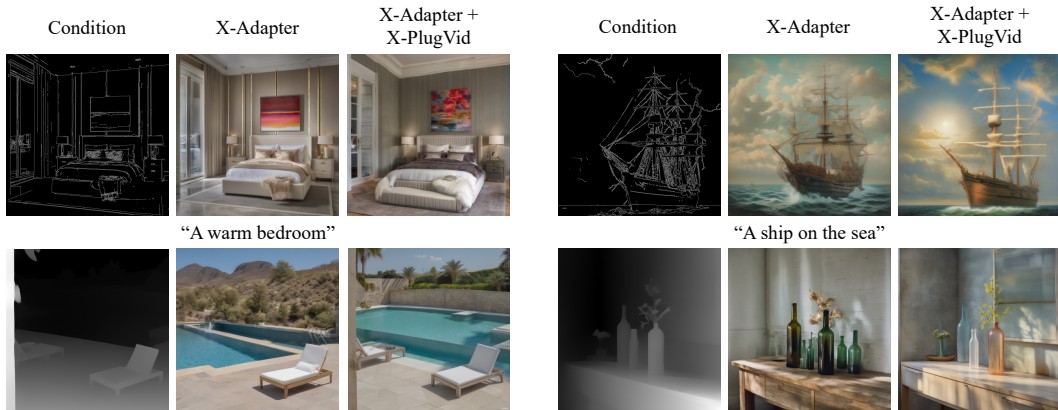

"A warm bedroom"                                    "A ship on the sea"

"There is a chair near the swimming pool"           "Several bottles on the table"

Figure 3: *Qualitative Result*. Compare to X-Adapter, our method improves its condition fidelity and image quality.

