# OpenReview forum: "X-PlugVid: Versatile Adaptation of Image Plugins for Controllable Video Generation"
_ICLR.cc/2025/Conference — ICLR 2025 Conference Withdrawn Submission_

### Official Review · Reviewer_fZpx · 2024-10-25

**Soundness:** 3
**Presentation:** 2
**Contribution:** 2
**Rating:** 5
**Confidence:** 4

**Summary:**

This paper (X-PlugVid ) proposes a spatial-temporal adapter that can make the video diffusion model compatible with existing image-based plug-and-play modules, e.g., ControlNet, T2I-adapter. X-PlugVid eliminates the needs to train a separated adapter for each ControlNet, rather it trains a universal adapter works for all types of ControlNets. The paper is heavily motived by X-Adapter which enables SDXL to use the ControlNets of SD 1.5. X-PlugVid extends the scope further so that the video diffusion model can use image ControlNets. Beside training a spatial-temporal adapter, this paper also introducing a new timestep remapping strategy to inject the high-frequency signals of image ControlNets into the video diffusion model to enhance video quality. Experiments on difference image condition plugins have demonstrated the effectiveness of the proposed method.

**Strengths:**

1. This paper tackles an important topic on how to leverage the rich image plugins (ControlNets, T2I-adapters) in video domain. Motived by the fact that finetuning for each plugin cannot scale, this paper proposes a universal spatial-temporal adapter for a family of plugins.

2. Compared to X-Adapter that works solely on image (SD1.5 -> SDXL), this paper wants to use image plugins in video diffusion. This paper has some observations on how ControlNet and X-Adapter works and then proposes some specific methodology, such as the spatial-temporal adapter and the new timestep remapping strategy.

**Weaknesses:**

- This paper only shows some quantitative metrics such as FID and Optical Flow Error, which are believed as incomprehensible when describing the quality of the generated videos. The human evaluation or user study is required to further verify the effectiveness of the method. For instance, the human evaluation can consider the temporal consistency and the overall visual quality.

- Further clarification is needed in Table 1 because all other competitors (ControlVideo, Control-A-Video, VideoComposer) are using image model as the base model while X-PlugVid is using a video model. It is expected that X-PlugVid can outperform in Table 1 because of a better base model. A good comparison would be X-PlugVid vs. other video ControlNets (starting from a video model, then add the spatial ControlNet, train the ControlNet and video model together like Control-A-Video).

**Questions:**

1. I am assuming that you are still use image depth/canny method to get the control sequences. How do you think of the fact that the control sequences are inconsistent? For instance, if you stack all the canny/depth control sequences together as a video, you may observe flickering between frames. Methods like Control-A-Video inflates the ControlNet branch into spatial-temporal to deal with this issue. What will X-PlugVid behave in this case?

---

### Official Review · Reviewer_xYJz · 2024-11-04

**Soundness:** 3
**Presentation:** 3
**Contribution:** 2
**Rating:** 5
**Confidence:** 4

**Summary:**

X-PlugVid is a framework designed to adapt pretrained image-based plug-and-play modules for use in video diffusion models, allowing controllable video generation. The framework uses a spatial-temporal adapter to bridge the gap between image and video diffusion, with a frozen image diffusion model (e.g., Stable Diffusion v1.5) providing spatial priors. A timestep remapping strategy injects information from later timesteps of the image model into earlier timesteps of the video model, enhancing quality and temporal consistency. Experimental results show X-PlugVid’s compatibility with various video models and image plugins, and its adaptability for controllable video generation, supported by ablation studies and qualitative results.

**Strengths:**

1. The use of an image model as a prior to support SVD for controllable video generation without additional training on either the video or image model is an interesting approach.
2. The paper is well-organized and easy to follow.

**Weaknesses:**

1. **Lack of Video Demonstrations**: There is no video demonstration provided to verify the effectiveness of the proposed method, which is a significant limitation for assessing its impact. Side-by-side comparisons with baselines, such as VideoComposer, ControlVideo, and Control-A-Video, are essential.

2. **Inference Time Concerns**: There is a concern regarding inference speed, as the image model seems to be involved at each inference step. Specifically, how long does it take to generate a 48-frame (2-second) video clip?

3. **Temporal Interpolation Challenge**: A key gap between controllable image generation and controllable video generation lies in producing reasonable interpolations when conditions vary significantly between frames. Appendix Table 2 lacks examples that address this challenge.

4. **Influence of Prompts**: The effect of prompts on the outcomes is not clearly analyzed, and there appears to be a lack of ablation experiments assessing this aspect.

**Questions:**

No questions

---

### Official Review · Reviewer_NDeZ · 2024-11-04

**Soundness:** 3
**Presentation:** 3
**Contribution:** 3
**Rating:** 3
**Confidence:** 5

**Summary:**

This paper introduces a framework named X-PlugVid, that leverages pre-trained image-based plugins for video generation without additional retraining. The proposed method includes a spatial-temporal adapter that transforms spatial control signals from image plugins (like ControlNet) into temporally coherent guidance for video diffusion models. The author proposes a timestep remapping strategy, which maps richer image features from later timesteps to guide early stages in video generation. The proposed X-PlugVid enhances controllable video generation, allowing a single adapter to make diverse plugins compatible with different video backbones.

**Strengths:**

1. The proposed training-free design enables seamless integration of image plugins into video models, saving computational resources.

2. The proposed adapter is compatible with various plugins and backbones, showing versatility across different video generation models.

3. By incorporating the timestep remapping strategy, the proposed method enables better control over the video generation process, enhancing guidance and improving video quality.

**Weaknesses:**

1. The model relies on high-frequency injection, which may introduce artifacts if not well-tuned, especially when aggressive controls are applied. The author should discuss this, as well as the failure cases.

2. The proposed high-pass filtering may limit performance with plugins that require low-frequency guidance. It would be great if the authors could discuss this in their paper/rebuttal.

3. The discussion of the proposed method in the introduction, and method sections are all around SVD. However, the main experiments are conducted on Hotshot-XL and I2VGen-XL. There are only a few discussions about applying the proposed method to SVD in the appendix which is weird. SVD is an image-to-video method and Hotshot-XL/I2VGen-XL are text/image-to-video models, there should be some difference between them when incorporating the proposed methods.

4. It is highly encouraged to upload the video by supplementary or to an anonymous account on YouTube/GitHub because not all the reviewers have a PDF reader that supports playing GIFs. Also, more video cases are encouraged. There are only 10 videos in the appendix. Please provide at least 20-30 video examples to better demonstrate the method's capabilities.

5. Is there any result that supports a wide-screen aspect ratio? For example, 16:9? Both Hotshot-XL and I2VGen-XL support this.

6. Why do the results in Table 2 in the Appendix have a text? The SVD method only supports image-to-video generation.

7. Table 2 in the Appendix should be Figure 2.

8. The name of UNet should be U-Net.

**Questions:**

Please refer to the weakness part.

My concerns were not addressed during the rebuttal, so I changed my score to reject.

---

### Note · Authors · 2024-12-12

I have read and agree with the venue's withdrawal policy on behalf of myself and my co-authors.